# Species specific morphological alterations in liver tissue after biliary occlusion in rat and mouse: Similar but different

**Beate Richter**[1]*, **Constanze Sänger**[1], **Franziska Mussbach**[1], **Hubert Scheuerlein**[2], **Utz Settmacher**[1], **Uta Dahmen**[1]

1 Department of General, Visceral and Vascular Surgery, University Jena, Jena, Germany, 2 Clinic for General, Visceral and Paediatric Surgery, St. Vincenz Hospital Paderborn, Teaching Hospital of the University Göttingen, Göttingen, Germany

* berichter@gmx.net

## Abstract

### Background

The selection of the appropriate species is one of the key issues in experimental medicine. Bile duct ligation is the mostly used experimental model in rodents to explore special aspects of occlusive cholestasis. We aimed to clarify if rats or mice are suitable for the same or different aspects in cholestasis research.

### Methods

We induced biliary occlusion by ligation and transection of the common bile duct (tBDT) in rats and mice (each n = 25). Recovery from surgical stress was assessed by daily scoring (stress score, body weight). At five different time points (days 1, 3, 7, 14, 28 after tBDT) we investigated hepatic morphometric and architectural alterations (Haematoxylin-Eosin staining, Elastica van Gieson staining) and the proliferative activities of parenchyma cells (Bromodeoxyuridine staining); as well as established systemic markers for liver synthesis, hepatocellular damage and renal dysfunction.

### Results

We found substantial differences regarding survival (rats: 100%, 25/25 vs. mice 92%, 22/25, p = 0.07) and body weight gain (p<0.05 at postoperative days 14 and 28 (POD)). Rats showed a faster and progressive hepatobiliary remodelling than mice (p<0.05 at POD 7+14+28), resulting in: i) stronger relative loss of hepatocellular mass (rats by 31% vs. mice by 15% until POD 28; p<0.05 at POD 7+14+28); ii) rapidly progressing liver fibrosis (p<0.05 at POD 14); iii) a faster and stronger proliferative response of parenchyma cells (hepatocytes: p<0.05 at POD 1+14+18; cholangiocytes: p<0.05 at POD 1+3+7+28); and iv) only tiny bile infarcts compared to mice (p<0.05 at POD 1+3+7+14). Both species showed comparable elevated markers of hepatocellular damage and serum bilirubin.

**Data Availability Statement:** All relevant data are within the paper.

**Funding:** This study was supported by the clinical research supporting program of the University of

Jena to B.R. ("IZKF-Rotationsprogramm", URL: http://www.izkf.uniklinikum-jena.de); and to U.D. by the German Federal Ministry for Education and Research (BMBF) Virtual Liver Network (URL: http://www.virtual-liver.de). The funders had no role in study design, data collection and analysis, decision to publish, or preparation of the manuscript.

**Competing interests:** The authors have declared that no competing interests exist.

## Conclusion

The key difference between rats and mice are the severity and dynamics of histological alterations, possibly accounting for their different susceptibilities for (septic) complications with low survival (mice).

## Introduction

The success of a project in experimental surgery is influenced by several major aspects: i) carefully defined scientific questions; ii) selection of the appropriate surgical model; iii) selection of the appropriate species; iv) knowledge regarding the species-specific effects on the characteristics of the "targeted human disease" [1, 2].

Selection of the appropriate species is often inspired by pragmatic reasons: costs and requirements of laboratory animal husbandry, experiences with handling of the species, available literature regarding the use of the surgical model in the specific species, and finally the skills of the research team. Since these pragmatic factors can be optimized, the knowledge-based selection of the appropriate species remains a challenge. The combination of the selected surgical model and the species needs to fulfil at least two requirements: resistance to surgical stress and the reliable development of the characteristics of the targeted human disease.

For decades, the experimental model of biliary occlusion has been used to mimic human cholestatic diseases to explore different aspects and potential mechanisms [2–11]. Mostly, rodents (either rats or mice) have been used, because of practical and scientific reasons. Projects with surgical aims tend to use rats because of the organ sizes, whereas projects with medical and pharmacological objectives favour mice for their abundance of molecular targets. Moreover, transgenic mice strains enable more detailed molecular research. Facing this challenge, we compared rats and mice after biliary occlusion regarding their similarities and differences in terms of resistance to surgical stress and hepato-biliary remodelling. We focussed on a limited number of essential parameters, that are investigated in different species in almost all related studies.

We intended to answer two specific questions with our study:

- Can we define relevant species-specific differences in hepatobiliary remodelling process after biliary occlusion?

- Do these differences imply any recommendation for the preferred use of either species in (surgical) cholestasis research?

## Materials and methods

### Experimental design

We performed the same experiment in two different species (rats and mice with each n = 25) in this study. All animals (n = 50) were subjected to triple ligation and transection of the ligated extrahepatic bile duct between the middle and distal ligature (tBDT) to induce total occlusive cholestasis. At five time points (day 1, 3, 7, 14, 28, each with n = 5 / time point) after tBDT, the animals were randomly assigned for sacrifice, and samples of blood and liver lobes were collected for further analyses.

## Animals

All surgical procedures were performed in inbred male mice (C57BL/6N, aged 9–10 weeks, body weight 25–28 g) or in inbred male rats (Lewis, aged 9–10 weeks, body weight 250–280 g). All animals were obtained from a commercial breeding laboratory (Charles River, Sulzfeld, Germany). The animals were fed a standard laboratory diet with water and (mouse or rat) chow ad libitum until harvest. All animals were kept under constant environmental conditions with a 12 h light–dark cycle in a conventional animal facility using environmentally enriched type IV cages in groups (2–3 rats; 2–3 mice). All procedures and housing of the animals were carried out according to the German Animal Welfare Legislation and approved by the local authorities (Landesamt für Verbraucherschutz Thüringen).

## Surgical technique

We induced biliary occlusion by triple ligation and transection of the ligated main extrahepatic bile duct (tBDT) between the middle and distal of three ligatures in 50 animals (n = 25 per species) as described before [12]. In detail, all interventions were performed at daytime under inhalation of isoflurane (mice: 1.5–2%, rats: 2.5–3% Isoflurane) mixed with pure oxygen at a flow of (mice: 0.3 L/min; rats: 0.5 L/min) (isoflurane vaporizer, Sigma Delta, UK) in a dedicated S1 operation room. At the end of the day the instruments were cleaned and sterilized in a commercial autoclave (Systec DE-23, Germany). All procedures were done under an operating microscope (Leica, magnification 10-25x, Germany) to ensure preservation of the branches of the hepatic artery and portal vein. All animals were weighed and then anaesthetized with isoflurane (mice: 2%; rats: 3%) and oxygen (mice: 0.3 L/min; rats: 0.5 L/min) in an induction chamber. The abdomen was shaved, the animals were placed in a supine position and the skin was disinfected with iodine solution. After transverse laparotomy, the distal part of the common extrahepatic bile duct below the bile ducts of the inferior liver lobes (caudate inferior lobe and right inferior lobe) was identified and separated from surrounding fat tissue. Special care was taken to avoid any injury of the pancreas tissue. Three ligatures were placed around the prepared segment of the common bile duct. The common bile duct was always transected between the middle and distal ligature. Closure of the abdominal wound was always done by two-layer running suture (Prolene 6–0, Ethicon) [12].

## Postoperative care and analgesic treatment of the animals

Analgesic treatment was started immediately after the wound closure in all animals. Buprenorphine (mice: 0.005 mg/kg BW; rats: 0.05 mg/kg BW, Temgesic®) was subcutaneously injected; the analgetic therapy was given twice a day during the first three postoperative days. During this time the animals were checked for their clinical condition twice per day; afterwards the animals were routinely checked once per day. Clinical scoring was performed according to Hawkins and GV-SOLAS [13, 14].

## Stress score and criteria for euthanasia of the animals

The stress score included four gradation according to Hawkins and GV-SOLAS [13, 14]: grade "0" brightness of eyes (irrespective of signs of icterus), normal behaviour, weight gain and food intake; grade "1" brightness of eyes (irrespective of signs of icterus), normal shining fur, weight gain, or weight loss < 5% / 24 h, food intake, not sitting in one edge of the cages, no hunching; grade "2" brightness of eyes (irrespective of signs of icterus), aggressive behaviour (against itself and other animals in cage), no hunching, normal shining fur, reduced food intake, weight loss ≤ 14% / 24 h; grade "3": dullness of eyes (irrespective of signs of icterus), untidy fur, nasal

discharge, apathy of the animal, persistent hunching even after analgetic treatment, weight loss of $\geq$ 15% / 24 h.

Since survival was an endpoint of the study, we defined criteria for euthanasia as follows: all animals showing a stress score of 3 were euthanized, as the condition of the animal was considered as moribund. For euthanasia the animal was anaesthetized with isoflurane (mice: 2%; rats: 3%) and oxygen (mice: 0.3 L/min; rats: 0.5 L/min) in an induction chamber and afterwards euthanized by cervical dislocation [13, 14].

A yellow colouring of the paws, urine or brightening faeces were accepted as signs of biliary occlusion ("icterus") and did not lead to euthanasia of the animal.

## Determination of the body weight gain, liver weights

The animals were daily weighed until the end of the observation period. The body weight gain was calculated by dividing the weight of the animal of the dedicated day [g] by the starting body weight [g] of the animal. The explanted liver was weighed using an analytical balance (BLC-3000, Germany). Liver body weight ratio was calculated by dividing the weight of the liver [g] by the starting body weight [g] of the animal, respectively.

We included the whole liver weight of an untreated male rat and mouse (representing "day zero") in our calculation for better understanding of the weight gain and ratio in either species (rat: 10 g, BW 250 g; and mouse 1.2 g, BW 25 g).

## Liver enzymes and systemic parameters

Serum was stored at -20˚C until measurement of the liver enzymes using an automated chemical analyser (Bayer Advia 1650, Germany).

## Histology and immunohistochemistry

Samples were taken from the middle part of every liver lobe assuring evaluation of comparable areas of the liver lobes in all animals. Sections, 4 μm thick, were cut after paraffin embedding.

Haematoxylin-Eosin staining (HE) was used for histologic and morphological analysis of the liver tissue; Elastica van Gieson (EvG) for quantification of relative content (relative area per slide) of collagen (Collagen Index) and for assessment of the distribution of fibrosis in relation to anatomical landmarks (Fibrosis Score); we used Bromodeoxyuridine (BrdU)-staining for detection of the proliferation indices of hepatocytes and cholangiocytes. Detailed descriptions of staining methods are listed in supplement. After staining all slides were digitalized using a slide scanner (Nanozoomer, Hamamatsu Electronic Press Co., Ltd, Lwata, Japan).

## Haematoxylin-Eosin staining (HE)

The samples were fixed in 4.5% buffered formalin for 48 h. Sections of 4 μm thickness were cut after paraffin embedding. Slides were stained with Haematoxylin-Eosin (HE) for histo-pathological examination. After staining, all slides were digitalized using a slide scanner (Nanozoomer 2.0 HT scanner and the software NDP.scan 2.3; Hamamatsu City, Japan).

Number and relative area of periportal fields as well as the biliary proliferates (diameter and number of bile ducts in periportal area) were evaluated with the measuring tool of NPG-Viewer ("NanoZoomer Digital Pathology"; Hamamatsu, Japan). Results were given as numerical value, for size in mm$^2$, and relative size in %. The relative area represents the area of ductular reaction in relation to area of the total section [%].

## Bromodeoxyuridine (BrdU staining)

The staining procedure was based on a modified protocol of Sigma Inc. After deparaffinization and rehydration, tissue sections were treated with prewarmed 0.1% trypsin solution at 37˚C for 20 minutes, followed by denaturation with 2 N HCl at 37˚C for 30 minutes, and blocking with avidin solution for 10 minutes, biotin solution for 10 minutes, and 5% goat serum BSA-TBS at 37˚C for 15 minutes. In the next step sections were incubated with 1:50 monoclonal anti-BrdU antibody (DAKO Inc.) at 37˚C for 1 hour, followed by 1:300 biotinylated Fab-specific goat anti-mouse linked antibody (Sigma Inc.) for 30 minutes and AP-conjugated streptavidin (DAKO Inc.) for 30 minutes, prior to the application of Neofuchsin solution for 20 minutes. The sections were washed, counterstained with Hematoxylin, and coverslipped with Immu-Mount (Shandon Inc.).

## Elastica-van-Gieson (EVG)

Formalin-fixed paraffin-embedded liver biopsy tissues were sectioned to a thickness of 4 μm and underwent Elastica van Gieson (EVG) staining using the following procedure: Deparaffinized and hydrated sections were dipped in 70% ethanol containing 1% hydrogen chloride, incubated in resorcin–fuchsin solution for 60 minutes, and washed in 100% ethanol and in water, followed by counterstaining with van Gieson's solution (saturated picric acid containing 0.09% acid fuchsin) for 5 minutes, and coverslipped with Immu-Mount (Shandon Inc.).

## Quantification of proliferation (BrdU)

The proliferative activity of hepatocytes (BrdU) and the quantification of accumulated fibrous tissue (Collagen-Index, EVG) were determined using the HistoKAt software developed at Fraunhofer MEVIS (Dr. Homeyer, Fraunhofer MEVIS, Bremen, Germany). This software can be trained to recognize certain structures (e.g., cell nuclei) or defined patterns and is suitable for batch analysis. The software was kindly provided by Fraunhofer-Institute (Fraunhofer MEVIS, Bremen, Germany) [15].

Proliferative activity of cholangiocytes was determined by counting BrdU-positive cholangiocytes per bile ducts in 10 HPF (40x magn.) of periportal fields and in 10 HPF of intralobular area ("extra-portal ductular reaction") per slide (using NPG-Viewer).

## Quantification of relative content of collagen and elastic fibres (Collagen-Index) and semi-quantitative assessment of the severity of fibrosis (Fibrosis score) using EVG staining

The Collagen Index was calculated irrespective of the location of the positively stained areas (periportal, pericentral).

To assess the severity of fibrosis, we additionally used the established fibrosis staging score according to Blunt modified for rodents by Lo and Gibson-Corley [16–18]. This score reflects

**Table 1. Modified fibrosis score according to Blunt, Lo and Gibson-Corley [17,18].**

| Score | Explanation |
|---|---|
| 0 | No fibrosis |
| 1 | periportal fibrosis |
| 2 | 1 + with pericentral fibrosis |
| 3 | 2 + with bridging fibrosis |
| 4 | cirrhosis |

location and extent of fibrosis and includes periportal, pericentral and bridging fibrosis and cirrhosis (see Table 1). We assessed 10 HPF (40x magn., EvG staining) of periportal and pericentral areas per slide and animal using the NPD-Viewer. The median of the fibrosis score is given to avoid under- or overscoring according to Lo and Gibson-Corley [17, 18].

### Statistical analysis

The data are expressed as mean ± standard deviation (SD) if not indicated otherwise. The data were analysed using SPSS (IBM SPSS 22 for Windows).

We did not include weight data in statistical analyses since rats presented always higher weights due to their greater body weight compared to mice, respectively. Therefore, we included only data expressing a relation (e.g., liver body weight ratio, body weight gain in %) and the data from histology and immunohistochemistry for statistical analyses.

Type of distribution was determined using the Kolmogorow-Smirnow test (including the correction of significance according to Lilliefors). As the tests revealed a non-normal distribution, the data were analysed using non-parametric tests (Kruskal-Wallis Test, Mann-Whitney-U-Test). Differences were considered significant if p-value of less than 0.05 (2-tailed) were obtained (NS: not significant).

## Results

### Tolerance to surgical stress was more pronounced in rats than in mice

**Survival and recovery of the animals.** All rats tolerated the procedure well throughout the planned observation period (100% survival, p = 0.07 vs. mice) without experiencing any complications. The rats showed a maximal weight loss of up to 9% within the first 3 postoperative days (p vs. mice), followed by a constant weight gain exceeding the starting weight within the first 7 days. Rats showed a significant stronger body weight gain at the late time points, POD 14 and POD 28 compared to mice (p<0.05, see Table 2).

Mice tolerated tBDT to a lower extent. Three mice (3/25; 92% survival) died before the intended sacrifice date: One mouse found dead in cage at POD 1, POD 3, POD 28, respectively. The autopsy excluded surgical complications. The surviving mice showed an initial weight loss until POD 3 by ~12% followed by a steady weight, albeit just reaching the starting weight within the observation time (see Table 3).

No animal was euthanized before the planned sacrifice date.

**Laboratory blood tests results.** As expected, BDL induced a cholestasis in both species, albeit following a different kinetic and severity. In rats, the total bilirubin in serum increased until reaching a constant plateau on POD 3. In contrast, in mice the level increased until POD 7 followed by milder increase thereafter.

In both species, the liver enzymes, indicative for hepatocellular damage, increased sharply at POD 1 and declined to persisting moderately elevated levels thereafter.

In contrast, albumin as a parameter of liver synthesis was slightly reduced. However, INR also indicative of the synthesis function of the liver, remained within the normal range. Kidney function was also not affected, with values within (rat) or below (mouse) normal range (see Tables 2 and 3).

**Liver weight gain.** In rats, we observed a steady increase in liver and spleen weight, resulting in a similar increase in the liver body weight ratio. In mice, liver weight and liver body weight ratio increased during the first week and remained stable thereafter. Rats showed a significantly stronger liver weight gain at POD 14 and 28 compared to mice (p<0.05). Mice showed a significant higher liver body weight ratio (lbwr) at the early time points, POD 1 and

**Table 2. Results of laboratory chemistry and data of survival and weight data after tBDT in rats.**

| | POD 1 | | | POD 3 | | | POD 7 | | | POD 14 | | | POD 28 | | |
|---|---|---|---|---|---|---|---|---|---|---|---|---|---|---|---|
| | mean | ± | STDV | mean | ± | STDV | mean | ± | STDV | mean | ± | STDV | mean | ± | STDV |
| **Laboratory Chemistry** | | | | | | | | | | | | | | | |
| ASAT [<0.83 µmol/l.s] | 16.83 | ± | 6.32 | 10.29 | ± | 1.92 | 7.12 | ± | 1.18 | 6.91 | ± | 1.59 | 8.38 | ± | 1.95 |
| ALAT [<0.74 µmol/l.s] | 12.00 | ± | 4.67 | 5.67 | ± | 1.63 | 2.33 | ± | 0.53 | 1.60 | ± | 0.38 | 3.66 | ± | 2.49 |
| Bilirubin (total) [<21 µmol/l] | 63.17 | ± | 7.54 | 150.25 | ± | 18.86 | 166.13 | ± | 27.84 | 171.40 | ± | 13.38 | 168.86 | ± | 20.51 |
| Albumin [33–53 g/l] | 6.17 | ± | 0.69 | 6.25 | ± | 0.43 | 6.50 | ± | 0.87 | 5.40 | ± | 0.49 | 5.60 | ± | 0.49 |
| Glucose [3.9–5.8 mmol/l] | 7.37 | ± | 0.87 | 6.13 | ± | 0.58 | 7.65 | ± | 1.05 | 6.86 | ± | 1.05 | 6.18 | ± | 1.21 |
| Creatinine [35–100 µmol/l] | 26.50 | ± | 2.93 | 28.50 | ± | 1.50 | 29.25 | ± | 1.85 | 29.80 | ± | 3.06 | 27.40 | ± | 1.36 |
| INR [71–120%] | 117.90 | ± | 2.81 | 118.27 | ± | 3.28 | 117.82 | ± | 4.01 | 119.01 | ± | 2.77 | 118.38 | ± | 1.35 |
| **Survival, weight data** | | | | | | | | | | | | | | | |
| survival at time point [number; %] | 5/5 | | 100% | 5/5 | | 100% | 5/5 | | 100% | 5/5 | | 100% | 5/5 | | 100% |
| stress score | 1.31 | ± | 0.48 | 1.79 | ± | 0.63 | 0.51 | ± | 0.32 | 0.24 | ± | 0.20 | 0.09 | ± | 0.10 |
| body weight [g] | 248 | ± | 12.39 | 231 | ± | 15.12 | 252 | ± | 11.07 | 264 | ± | 14.76 | 269.59 | ± | 15.06 |
| body weight gain [%] | 97.39 | ± | 2.01 | 90.66 | ± | 2.86 | 98.91 | ± | 3.21 | 103.61 | ± | 1.98 | 105.72 | ± | 2.37 |
| liver weight [g] | 11.85 | ± | 0.81 | 14.30 | ± | 0.45 | 16.25 | ± | 1.67 | 17.76 | ± | 2.97 | 18.93 | ± | 2.29 |
| liver weight gain [%] | 118.5 | ± | 2.32 | 143 | ± | 2.18 | 162.5 | ± | 2.91 | 177.58 | ± | 1.98 | 189.3 | ± | 3.01 |
| liver body weight ratio [%] | 3.70 | ± | 0.32 | 4.17 | ± | 0.27 | 5.59 | ± | 0.32 | 6.39 | ± | 1.11 | 6.81 | ± | 0.78 |
| spleen weight [g] | 0.52 | ± | 0.03 | 0.58 | ± | 0.06 | 0.87 | ± | 0.18 | 0.73 | ± | 0.23 | 1.06 | ± | 0.21 |

POD 3 (p<0.05), compared to rats. Whereas rats showed again a significantly increased lbwr at POD 28, compared to mice (p < 0.05 (see Fig 1, Tables 2 and 3).

## Hepatobiliary remodelling in rats is significantly stronger compared to mice

**Histology (HE) and immunohistochemistry (BrdU, EvG).** In both species hepatobiliary remodelling occurred in response to the biliary occlusion. The ductular reaction due to tBDT

**Table 3. Results of laboratory chemistry and data of survival and weight data after tBDT in mice.**

| | POD 1 | | | POD 3 | | | POD 7 | | | POD 14 | | | POD 28 | | |
|---|---|---|---|---|---|---|---|---|---|---|---|---|---|---|---|
| | mean | ± | STDV | mean | ± | STDV | mean | ± | STDV | mean | ± | STDV | mean | ± | STDV |
| **Laboratory Chemistry** | | | | | | | | | | | | | | | |
| ASAT [<0.83 µmol/l.s] | 53.93 | ± | 26.50 | 13.25 | ± | 3.32 | 19.01 | ± | 10.41 | 17.68 | ± | 10.97 | 11.14 | ± | 2.92 |
| ALAT [<0.74 µmol/l.s] | 29.24 | ± | 9.35 | 10.68 | ± | 1.78 | 11.11 | ± | 2.45 | 9.23 | ± | 5.18 | 7.94 | ± | 2.12 |
| Bilirubin (total) [<21 µmol/l] | 127.83 | ± | 57.94 | 166.50 | ± | 49.32 | 187.98 | ± | 30.18 | 190.89 | ± | 84.46 | 199.01 | ± | 41.55 |
| Albumin [33–53 g/l] | 9.50 | ± | 1.71 | 8.83 | ± | 0.37 | 9.40 | ± | 0.80 | 9.50 | ± | 0.50 | 8.00 | ± | 0.63 |
| Glucose [3.9–5.8 mmol/l] | 8.63 | ± | 0.48 | 8.30 | ± | 0.95 | 9.12 | ± | 0.80 | 7.10 | ± | 1.88 | 8.50 | ± | 2.12 |
| Creatinine [35–100 µmol/l] | 24.00 | ± | 5.89 | 21.00 | ± | 10.17 | 33.20 | ± | 4.35 | 18.38 | ± | 11.44 | 38.25 | ± | 6.92 |
| INR [71–120%] | 115.12 | ± | 1.57 | 116.86 | ± | 1.89 | 114.92 | ± | 3.28 | 118.89 | ± | 2.95 | 119.83 | ± | 3.01 |
| **Survival, weight data** | | | | | | | | | | | | | | | |
| survival at time point [number; %] | 4/5 | | 80% | 4/5 | | 80% | 5/5 | | 100% | 5/5 | | 100% | 4/5 | | 80% |
| stress score | 1.73 | ± | 0.51 | 2.20 | ± | 0.65 | 1.83 | ± | 0.48 | 0.91 | ± | 0.27 | 0.32 | ± | 0.13 |
| body weight [g] | 26.81 | ± | 0.70 | 25.25 | ± | 0.66 | 25.98 | ± | 0.12 | 26.61 | ± | 1.61 | 26.97 | ± | 1.05 |
| body weight gain [%] | 93.15 | ± | 2.53 | 87.73 | ± | 3.33 | 90.25 | ± | 2.21 | 92.44 | ± | 8.84 | 93.69 | ± | 2.14 |
| liver weight [g] | 1.36 | ± | 0.06 | 1.78 | ± | 0.16 | 1.81 | ± | 0.33 | 1.74 | ± | 0.42 | 1.79 | ± | 0.03 |
| liver weight gain [%] | 112.92 | ± | 1.05 | 148.33 | ± | 1.33 | 150.83 | ± | 1.41 | 144.64 | ± | 1.21 | 148.93 | ± | 1.01 |
| liver body weight ratio [%] | 4.69 | ± | 0.08 | 5.68 | ± | 0.09 | 5.80 | ± | 0.51 | 6.01 | ± | 0.38 | 6.06 | ± | 0.12 |
| spleen weight [g] | 0.06 | ± | 0.01 | 0.13 | ± | 0.04 | 0.15 | ± | 0.02 | 0.15 | ± | 0.06 | 0.13 | ± | 0.02 |

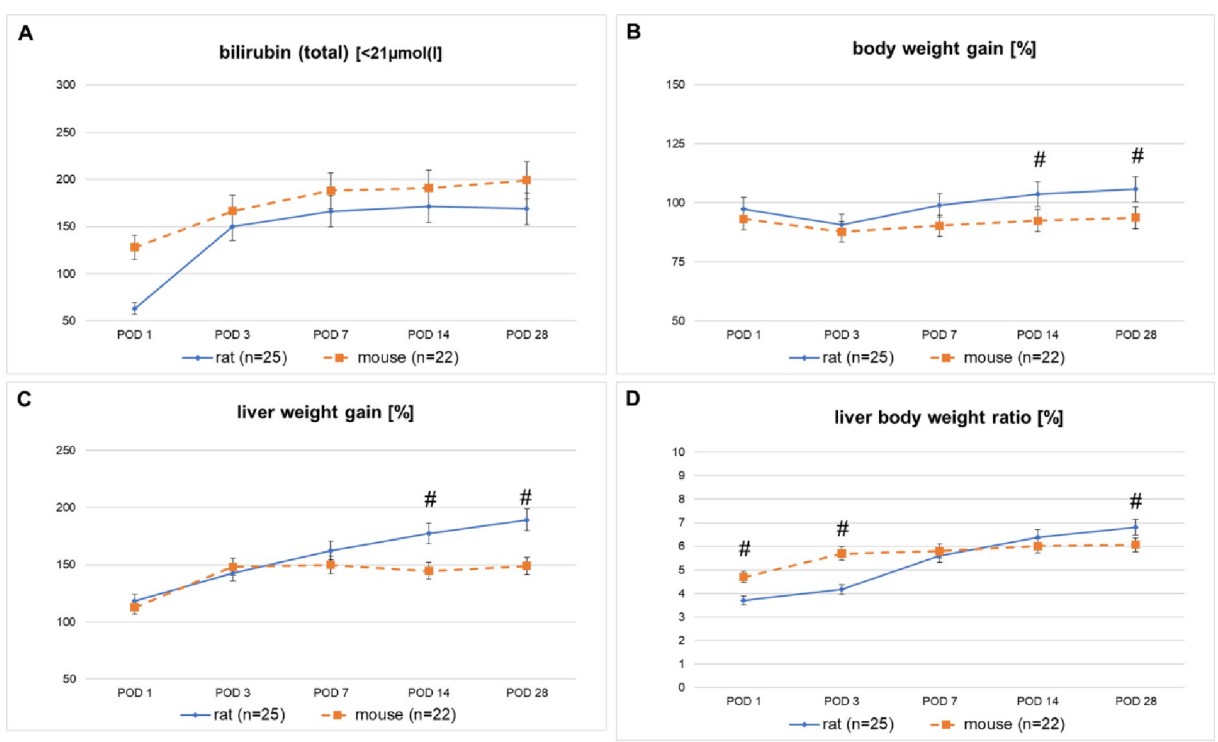

**Fig 1. A-D:** Results of bilirubin (total) in serum, and the liver and body weight gain with the related liver body weight ratio of rats and mice after tBDT.

led to an enlargement of the portal fields and of the biliary proliferates in the hepatocellular compartment. Morphometric analysis revealed a relative reduction of the hepatocellular compartment due to the relative increase of the biliary compartment (see Figs 2 and 3, Tables 2 and 3).

**Morphometric analysis revealed substantial inter-species differences in the dynamics and extent of the hepatobiliary remodelling.** In rats, the hepatocellular compartment was reduced by ~31% within the observation time of 28 days, in mice only by ~15% ($p < 0.05$ at POD 7, 14, 28, respectively).

The differences resulted predominantly from significantly different dynamics of the enlargement of the cholangiocytes'compartment ($p < 0.05$ at POD 7, 14, 28, respectively). Rats showed a constant and significant stronger increase of the ductular reaction in the portal field and in the hepatocellular compartment (biliary proliferates), compared to mice (see Figs 2, 3 and 6; Tables 4 and 5).

In mice, we found a two-phased time course of hepato-biliary remodelling. We found an almost simultaneous increase in ductular reaction and formation of necrotic areas until POD 3, followed by a further enlargement of the portal fields and a decrease of necrotic area until POD 7, whereas at the late time points POD 14 and POD 28 we found a stable extension of the portal fields with simultaneously weaker increase of the biliary proliferates area (see Figs 2, 3 and 6; Tables 4 and 5).

There were also striking differences in respect to the extent of confluent necrosis. Rats developed small negligible areas of peribiliary necrosis. In mice, we found significantly more ($p < 0.05$ at all time-points, respectively) and larger confluent necrosis ($p < 0.05$ at all time-points, respectively) around the biliary proliferates developed with a maximum observed on POD 3 after tBDT (see Fig 3).

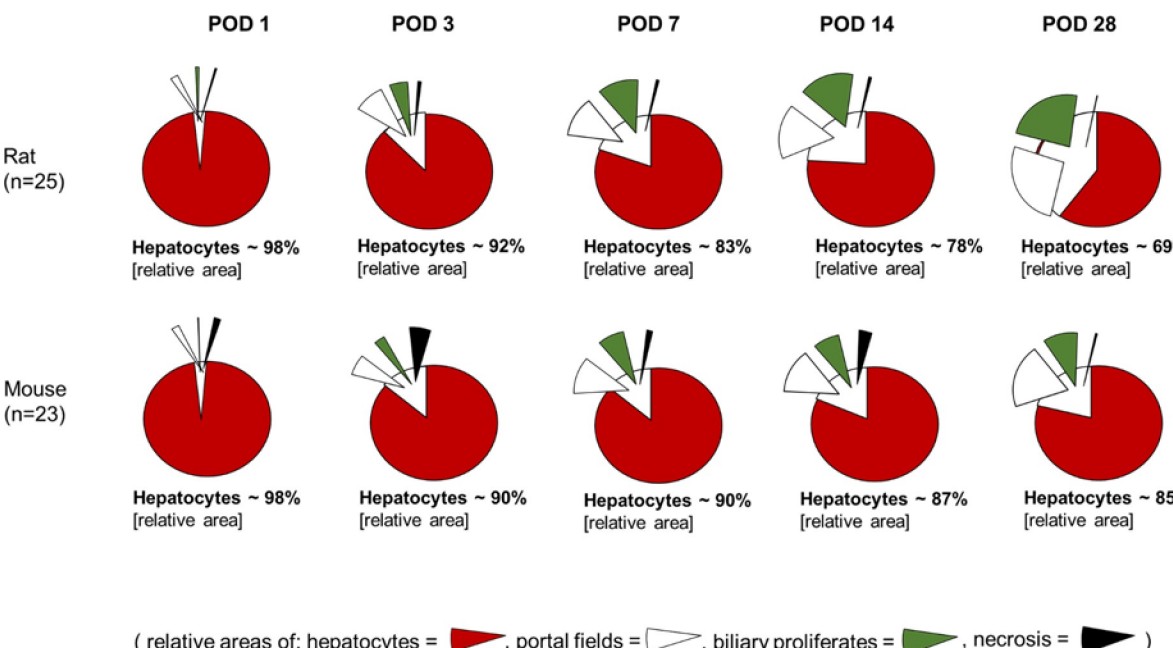

**Fig 2. Morphometric alterations of liver tissue compartments (using relative areas of hepatocytes, portal fields, biliary proliferates and necrotic area) in rat and mouse at different time points after tBDT.**

**Hepatobiliary proliferative activity in rats is significantly more pronounced than in mice.** In both species considerable proliferative activity of hepatocytes and cholangiocytes was observed but peaking at different time points. Hepatocytes'proliferation was significantly stronger in rats at almost all time points, reaching a maximum of 10% at POD 14 (p<0.05 at POD 1, 14,28, respectively). In mice, the peak proliferation reached only 5%, but occurred earlier at POD 7 (see Figs 4 and 6; Tables 4 and 5).

The differences in cholangiocytes'proliferation were even more striking. In rats, the significantly stronger proliferative rate of about 22% was already observed on POD 3 (p<0.05 at POD 1, 3,7, 28, respectively) and remained in this range throughout the first postoperative week, before declining gradually to 10% on POD 28. In mice, the peak proliferate rate reached only about 15% and was observed on POD 3 and decreased to 5% on POD 28 (see Figs 4 and 6; Tables 4 and 5, p < 0.05).

## Periportal liver fibrosis develops faster in rats than in mice

Time courses and extent of collagen deposits were also different in both species. In rats, a steady increase was observed and reached the maximum at 28 days (p<0.05 at POD14). In mice, the maximum was also reached at the end of the observation time but followed a slower progress with the maximal increase in the last week (see Fig 5A). We determined a significant stronger Collagen Index in rats at POD 14 compared to mice (p<0.05). At the other time points we found substantial but non-significant higher Collagen Indices in rats compared to mice.

The fibrosis score revealed the differences in the distribution in addition to the severity. In rats, the transition from periportal to bridging fibrosis occurred between POD 7 and 14; in mice later between POD 14 to 28 (see Figs 5 and 6; Tables 4 and 5). We detected a significantly stronger fibrosis score in rats only at POD 14 compared to mice (see Fig 5A and 5B, p < 0.05) staining.

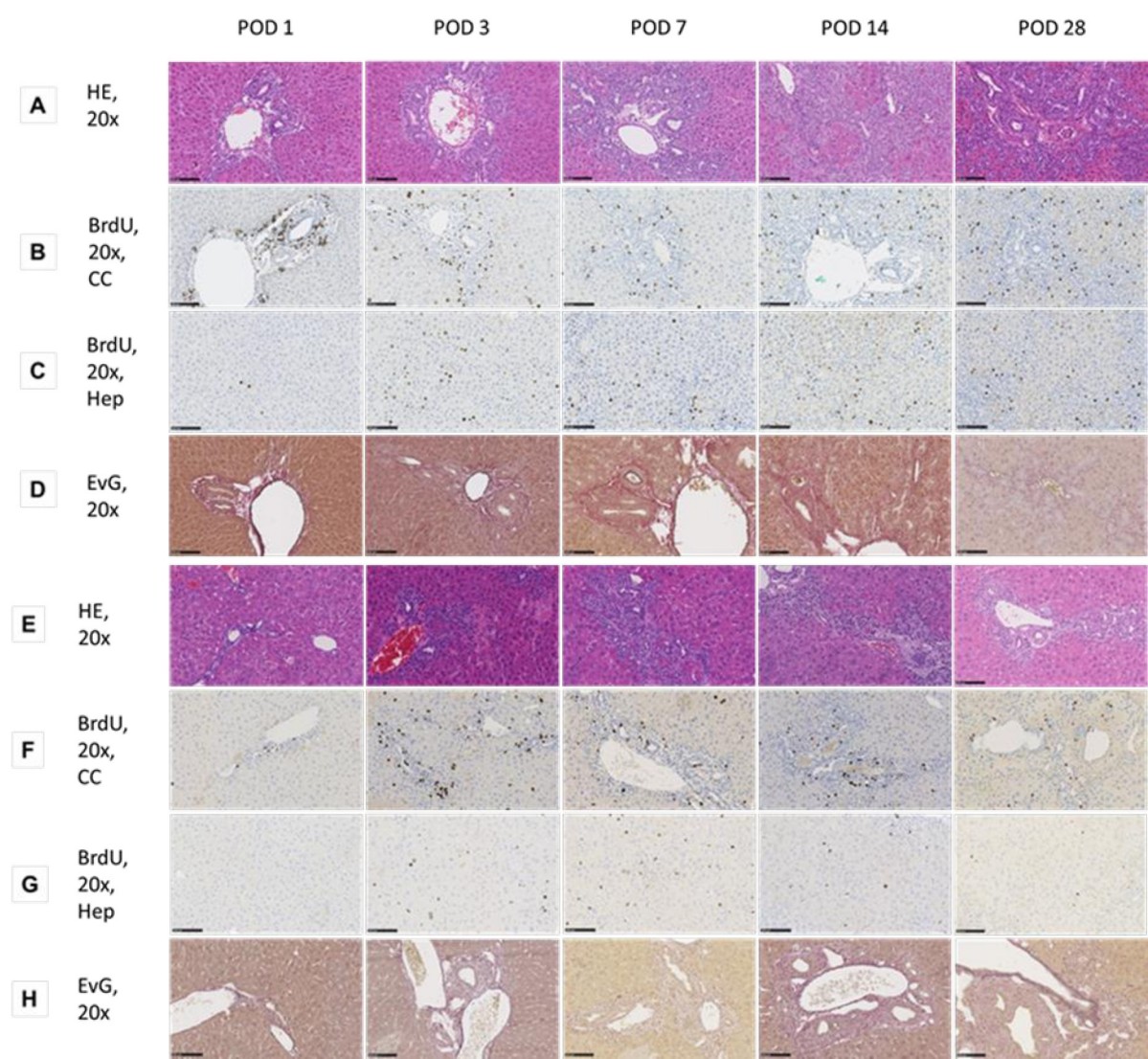

**Fig 3. A-H:** Histological and immunohistochemical images (HE, BrdU, EVG) after tBDT in rats (A-D) and mice (E-H).

## Discussion

### Resistance to surgical stress

The literature describes two reliable parameters to assess resistance to surgical stress in experimental surgery: Survival rate as the ultimate hard criteria and body weight gain as the strongest parameter to judge the condition of the surviving animals [13, 14, 19]. In our study, the rats showed a superior resistance indicated by the survival of all animals experiencing only minor weight loss of less than 10%. In contrast, some mice died during the observation period resulting in a survival rate of 92% and experienced a substantial weight loss preventing the recovery to the starting body weight within the observation period of 28 days. The distinct differences of the survival rates were not statistical significant, maybe mostly related to a small group size of 5 animals per time-point and species. However, our data showed a clear tendency for a better survival and stronger robustness of rats after tBDT. The literature provides only limited data regarding survival rate and causes of death after biliary occlusion in rodents. The reported

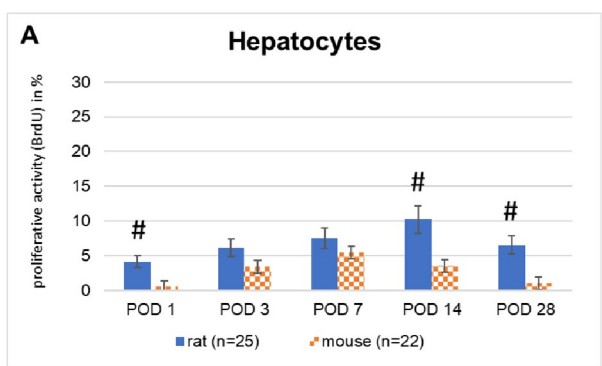
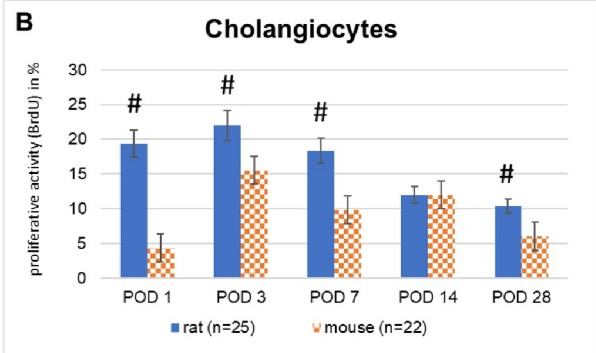

**Fig 4. A, B:** Proliferative activity of hepatocytes and cholangiocytes in rats and mice after total biliary occlusion.

survival data range from 60–97% [20–28]. In our study, no signs of surgical complication (e.g., bleeding, ischemia, biliary leakage), were identified during autopsy. Some authors concluded that mice are not suitable as animal model of biliary occlusion due the high perioperative mortality and the susceptibility to complications [2, 3, 26, 28]. However, in the three mice dying spontaneously, the number of visible liver necroses was higher than in all other mice sacrificed at the designated time points. Therefore, we attributed the death of the animals to complications due to tBDT. A current study in mice after BDL revealed so-called bile infarcts occurring by rupture of the apical membrane of hepatocytes. The rupture of the apical membrane of hepatocytes can lead to single-cell bile microinfarcts, or via a domino-effect to necrosis of multiple hepatocytes resulting in large bile infarcts [29]. In addition, the authors found no

**Table 4. Morphology and results of immunohistochemistry (HE, BrdU, EVG) after tBDT in rats.**

| | POD 1 | | | POD 3 | | | POD 7 | | | POD 14 | | | POD 28 | | |
|---|---|---|---|---|---|---|---|---|---|---|---|---|---|---|---|
| | mean | ± | STDV | mean | ± | STDV | mean | ± | STDV | mean | ± | STDV | mean | ± | STDV |
| **Portal fields (PF)** | | | | | | | | | | | | | | | |
| relative area of portal fields [%] | 1.76 | ± | 1.21 | 5.69 | ± | 1.37 | 8.29 | ± | 1.03 | 11.34 | ± | 2.79 | 16.87 | ± | 2.42 |
| number of portal fields | 13.29 | ± | 2.85 | 15.03 | ± | 1.51 | 14.00 | ± | 3.78 | 14.16 | ± | 2.58 | 13.57 | ± | 3.62 |
| number of bd per PF | 7.35 | ± | 2.31 | 22.38 | ± | 7.35 | 29.13 | ± | 2.81 | 35.05 | ± | 9.50 | 80.31 | ± | 6.78 |
| diameter of bd per PF [μm] | 5.30 | ± | 1.32 | 16.54 | ± | 3.24 | 20.09 | ± | 3.68 | 29.80 | ± | 5.14 | 33.18 | ± | 4.67 |
| **Extraportal ductular reaction** | | | | | | | | | | | | | | | |
| relative area [%] | 0.25 | ± | 0.72 | 2.26 | ± | 0.98 | 8.61 | ± | 1.21 | 10.68 | ± | 1.67 | 14.21 | ± | 2.53 |
| number of biliary convolutes | 5.38 | ± | 1.32 | 12.75 | ± | 2.31 | 14.85 | ± | 3.11 | 15.74 | ± | 7.07 | 21.09 | ± | 8.31 |
| number of BD per convolute | 3.4 | ± | 1.31 | 13.98 | ± | 1.26 | 51.09 | ± | 2.01 | 65.3 | ± | 6.56 | 88.91 | ± | 5.89 |
| diameter of bd per convolute [μm] | 2.56 | ± | 1.57 | 12.58 | ± | 1.83 | 16.13 | ± | 3.21 | 18.31 | ± | 5.78 | 21.45 | ± | 3.94 |
| **Hepatocytes** | | | | | | | | | | | | | | | |
| relative area [%] | 97.98 | ± | 1.25 | 92.04 | ± | 1.93 | 82.94 | ± | 1.73 | 77.83 | ± | 1.64 | 68.92 | ± | 3.64 |
| **Necrosis** | | | | | | | | | | | | | | | |
| number | 0.3 | ± | 0.72 | 0.41 | ± | 0.53 | 1.69 | ± | 0.78 | 1.18 | ± | 0.25 | 0 | ± | 0 |
| relative area [%] | 0.01 | ± | 0.51 | 0.01 | ± | 0.31 | 0.16 | ± | 0.42 | 0.15 | ± | 0.36 | 0 | ± | 0 |
| **Proliferation-Index (BrdU)** | | | | | | | | | | | | | | | |
| hepatocytes | 4.13 | ± | 2.82 | 6.12 | ± | 2.39 | 7.54 | ± | 2.81 | 10.2 | ± | 3.8 | 6.52 | ± | 3.17 |
| cholangiocytes | 19.34 | ± | 6.87 | 22.01 | ± | 7.37 | 18.31 | ± | 2.93 | 12.00 | ± | 3.21 | 10.39 | ± | 3.28 |
| **EvG** | | | | | | | | | | | | | | | |
| Collagen-Index | 6.45 | ± | 1.76 | 9.59 | ± | 1.89 | 12.32 | ± | 2.01 | 19.37 | ± | 3.97 | 27.66 | ± | 9.51 |
| Fibrosis score | 0.75 | ± | 0.43 | 0.917 | ± | 0.28 | 1.167 | ± | 0.55 | 2.167 | ± | 0.99 | 2.833 | ± | 0.55 |

**Table 5. Morphology and results of immunohistochemistry (HE, BrdU, EVG) after tBDT in mice.**

| | POD 1 | | | POD 3 | | | POD 7 | | | POD 14 | | | POD 28 | | |
|---|---|---|---|---|---|---|---|---|---|---|---|---|---|---|---|
| | mean | ± | STDV | mean | ± | STDV | mean | ± | STDV | mean | ± | STDV | mean | ± | STDV |
| **Portal fields (PF)** | | | | | | | | | | | | | | | |
| relative area of portal fields [%] | 1.02 | ± | 0.57 | 3.92 | ± | 1.27 | 6.34 | ± | 2.01 | 7.12 | ± | 3.25 | 11.56 | ± | 6.14 |
| number of portal fields | 12.36 | ± | 2.49 | 13.03 | ± | 2.84 | 14.35 | ± | 2.87 | 14.78 | ± | 4.31 | 15.01 | ± | 5.58 |
| number of bd per PF | 2.67 | ± | 0.98 | 7.98 | ± | 1.31 | 11.65 | ± | 2.57 | 14.26 | ± | 4.62 | 18.67 | ± | 4.38 |
| diameter of bd per PF [μm] | 5.34 | ± | 1.34 | 9.87 | ± | 2.64 | 10.42 | ± | 4.81 | 9.34 | ± | 3.56 | 11.85 | ± | 3.31 |
| **Extraportal ductular reaction** | | | | | | | | | | | | | | | |
| relative area [%] | 0.01 | ± | 0.34 | 0.89 | ± | 0.64 | 2.31 | ± | 0.93 | 2.87 | ± | 1.12 | 3.02 | ± | 1.21 |
| number of biliary convolutes | 3.00 | ± | 0.21 | 6.35 | ± | 1.82 | 15.83 | ± | 3.25 | 21.45 | ± | 5.45 | 24.53 | ± | 6.51 |
| number of bd per convolute | 2.45 | ± | 0.78 | 2.76 | ± | 1.56 | 4.57 | ± | 1.03 | 4.37 | ± | 1.73 | 5.31 | ± | 1.83 |
| diameter of bd per convolute [μm] | 4.87 | ± | 1.67 | 6.12 | ± | 3.81 | 6.75 | ± | 2.73 | 8.98 | ± | 2.91 | 9.01 | ± | 3.01 |
| **Hepatocytes** | | | | | | | | | | | | | | | |
| relative area [%] | 97.99 | ± | 1.72 | 89.99 | ± | 5.24 | 89.85 | ± | 3.76 | 86.88 | ± | 4.35 | 85.21 | ± | 5.01 |
| **Necrosis** | | | | | | | | | | | | | | | |
| number | 10.67 | ± | 3.89 | 17.71 | ± | 5.27 | 7.12 | ± | 2.32 | 17.69 | ± | 7.35 | 5.35 | ± | 1.28 |
| relative area [%] | 0.98 | ± | 0.52 | 5.2 | ± | 1.84 | 1.5 | ± | 0.49 | 3.13 | ± | 1.45 | 0.21 | ± | 0.34 |
| **Proliferation-Index (BrdU)** | | | | | | | | | | | | | | | |
| hepatocytes | 0.50 | ± | 0.32 | 3.41 | ± | 0.82 | 5.43 | ± | 0.65 | 3.52 | ± | 1.01 | 1.03 | ± | 0.74 |
| cholangiocytes | 4.36 | ± | 0.56 | 15.52 | ± | 2.71 | 9.82 | ± | 1.83 | 11.98 | ± | 2.58 | 6.03 | ± | 1.36 |
| **EvG** | | | | | | | | | | | | | | | |
| Collagen-Index | 5.49 | ± | 1.41 | 6.38 | ± | 1.19 | 7.34 | ± | 2.37 | 10.43 | ± | 3.03 | 23.68 | ± | 6.47 |
| Fibrosis score | 0.25 | ± | 0.43 | 1.11 | ± | 0.31 | 1.33 | ± | 0.43 | 1.44 | ± | 0.71 | 1.78 | ± | 0.43 |

evidence for intrahepatic perfusion disturbances resulting in a classic ischemia in their study. Interestingly, the term bile infarcts was first used in 1887 and described a complication of cholestasis in humans: intraparenchymal bile leakage leading to hepatocytes'necrosis. The term was first described by Jean-Martin Charcot and Albert Gombault and afterwards named as "Charcot-Gombault necrosis" [29]. However, one experimental study reports about formation of bile infarcts in rats within the acute phase after ligation of the common bile duct (cBDL) [30]. The authors focussed on the first 30 h after cBDL and found periportal bile infarcts already 6 h after cBDL without further increase in area until 30 h (= end of observation period). In addition, their rats showed a stronger elevation of transaminases and a reduced

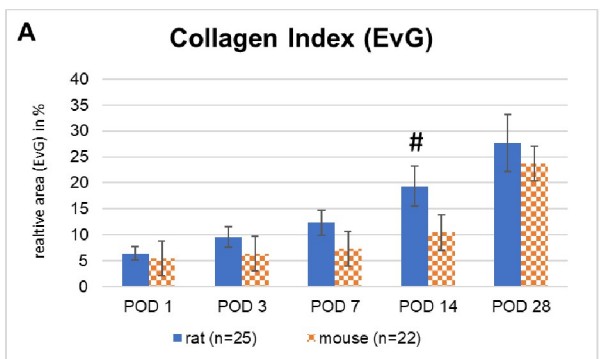
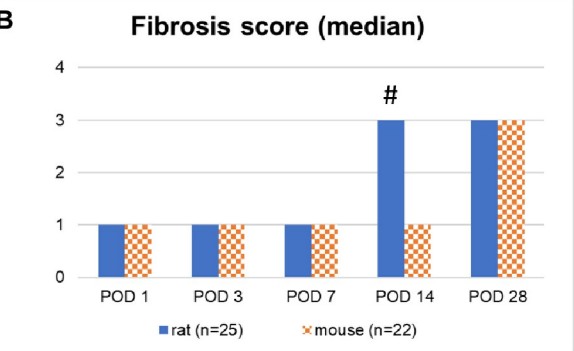

**Fig 5. A, B:** Relative area of collagen (Collagen Index) and zonal distribution of fibrous tissue (Fibrosis score) in rats and mice after tBDT (EvG staining).

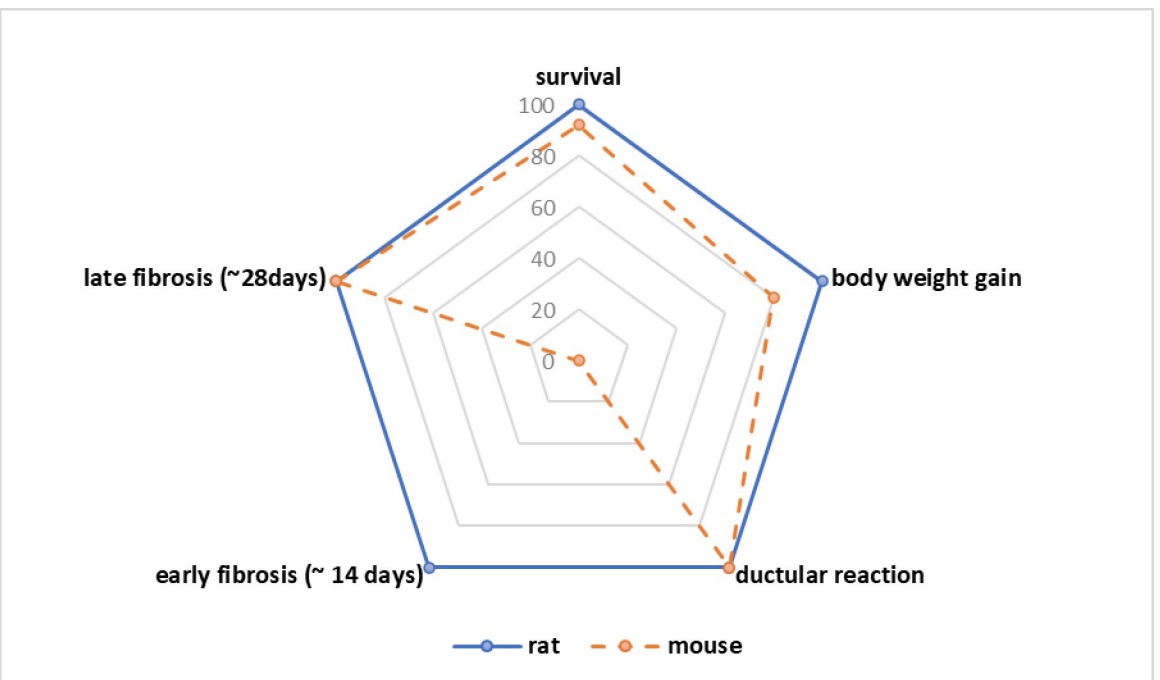

**Fig 6. Comparison of the differences in essential parameters for selecting the appropriate species (rat vs. mouse) in cholestatic research.**

survival of 65% (13/20) until 30 h after cBDL than our rats, respectively. In contrast, we found a different time course of bile infarcts in rats after tBDT, with slow progression of small necroses peaking on day 7, followed by rather fast regression until day 28 (= end of observation period). However, the literature provides no explanation for our striking differences in formation of necrosis/ bile infarcts after tBDT between rats and mice. Assuming the same mechanism for bile infarcts in rodents, a best potential explanation for the differences might be the (organ) size differences between rats and mice. Since mice have ca. 10% of the rats' size (e.g., liver weight, body weight, blood volume), such a micro structured organism mice might react extremely more sensitive to changes (e.g., intraductal pressure, surgical stress, bleeding) than a macro structured organism rat (see Table 6). Furthermore, considering size as a relevant factor in experimental surgical research, it is still noteworthy, that the anatomical proportions in mice demand for very profound experiences in hepato-biliary microsurgery [31].

### Anatomical differences

Focussing on the liver, four main differences between rodents and humans are well known: i) the absence of a gall bladder in rats; ii) the distinct lobulation of the rodent liver; iii) the increased liver body weight ratio in rodents; iv) and the increasing arterial blood supply with increasing liver size among the mammals [2, 3, 8, 28, 32] (see Table 6). Regarding the absent gall bladder in rats, the literature provides no information about the relevance for cholestasis research yet. The latter three are critical issues in surgical research and projects addressing the hepatic regenerative capacity, especially when subjecting the animal to repeated regeneration stimuli as in multi-staged hepatectomy [32–36]. In our study, we observed the characteristic responses to biliary occlusion in both species: the ductular reaction and a persistently increased bilirubin (total) level in serum.

**Table 6. Summary of characteristics of either species regarding hepatobiliary modelling after tBDT.**

| Characteristics | Special features | Mouse | Rat | Human (biliary occlusive diseases: e.g., extra and intrahepatic tumours) |
|---|---|---|---|---|
| Size | Body weight<br>liver weight (untreated liver)<br>Cholestatic liver weight (tBDT):<br>1 week | 25-30g<br>~ 1.2g<br>(untreated male C57BL/6N mouse, BW 25g)<br>~ 1.5g | 250-300g<br>10g<br>(untreated male lewis rat, BW 250g)<br>~ 16g | 70-100kg<br>~ 1.8kg<br>(healthy male, BW 75kg)<br>no data |
|  | 2 weeks<br>4 weeks | ~ 1.7g<br>~ 1.8g | ~ 18g<br>~ 19g |  |
| Costs |  | depending on age, weight, genetic background of the animals and the pricing of the supplier<br>+ costs for the animal husbandry | | not applicable |
| Anatomical differences | gall bladder<br>Liver lobules | present<br>4 lobes | absent<br>4 lobes | present<br>(if not already removed for benign reasons)<br>2 lobes |
| Tolerance to surgical stress |  | low/ moderate | high | low-moderate-high<br>(depending on patient's characteristics,<br>pre-existing "liver function" and<br>liver diseases (e.g., hepatitis, PBC), and tumour localisation) |
| Susceptibility to complications |  | high | low | low-moderate-high<br>(depending on patients'characteristics: see above) |
| Kinetic and characteristics of hepatobiliary remodelling (assuming tBDT in rodents) | blood bilirubin levels<br>ductular reaction<br>Replacement of HC compartment (%, peak)<br>Fibrosis<br>(score, peak) | stable elevated<br>present<br>(moderate, slow progression)<br>slow progression<br>(~13%, POD 14)<br>(~15%, POD 28)<br>slow progression<br>(1, POD 14)<br>(3, POD 28) | stable elevated<br>present<br>(strong, rapid progression)<br>rapid progression<br>(~22%, POD 14)<br>(~30%,POD 28)<br>rapid progression<br>(3, POD 14)<br>(3, POD 28) | Dynamics and extent of cholestatic alterations (e.g., blood tests, liver tissue) depend on localisation of tumour (biliary occlusion) and<br>pre-existent liver diseases (e.g., hepatitis, PBC) |
| Knowledge about genetic models or genetic background of cholestatic diseases |  | high | low | not applicable (maybe the epigenetic background might be one of the future topics to address in research) |

## Dynamics of hepatobiliary remodelling

However, we found impressive and significant differences in the dynamics and the extent of the hepato-biliary remodelling. The rats showed a rapid remodelling of the liver architecture with expansion of portal fields and partial replacement of the hepatocellular compartment by biliary proliferates as well as the development of hepatic fibrosis. Despite principally similar characteristics, the dynamic of these alterations was different in mice. Mice showed a comparable distorted liver architecture much later—after four weeks. The literature describes in part similar results predominantly for rats, whereas we found more studies with divergent results in mice [26–28, 33]. The literature provides no explanation for the differences between the studies and species (see Table 6). Interestingly, in mice the genetic background seems to have a greater impact on the development of fibrosis, compared to rats [37].

## Limitations of our study

Our study contains some limitations. One challenge of our study was to focus on a limited number of essential parameters assuring their reliable comparison in either species. In

planning of the study we found that only the male animals and almost two strains of either species (without any genetical modifications) were used. Furthermore, a broad spectrum of genetically modified mice strains were used to address various specialised pathophysiological, pharmacological and molecular topics. In order to avoid a never ending project, we used only the male animals of one strain without genetical modifications per species and limited the study design on three endpoints. The strains were inbred C57BL/6N mice and inbred Lewis rats [2, 28, 37]. The endpoints were survival, stress resilience (e.g., body weight gain, stress score) and the hepato-biliary remodelling after tBDT. In retrospect, another limitation of the study could be the use of only male animals. Since the literature provides mainly studies using male animals in experimental research (irrespective of the species and the topic in cholestasis research), we decided to include only male animals in our project. To date, a discussion raised in the research community about using also female animals in order to balance the sex ratio in experimental research. Finally, all limitations harbour the risk of challenging results. However, our intention was not to summarize the knowledge of cholestasis research in mice and rats.

We wanted to create a robust data basis enabling fast and reliable decision for either species in relation to the end points of the project (e.g., timing of investigation).

## Genetic models

Our intention was to compare the most frequently used strains of rats and mice regarding their differences and similarities in hepatobiliary remodelling after biliary occlusion. Since genetically modified mice are primarily used for highly specialized projects, we focussed on projects without genetically modified rodents. The literature provides no data regarding the impact of certain genetically modified strains of either species on the basic characteristics of biliary occlusion. Finally, we restrained from including genetically modified mice strains, since we have not found equivalent genetically modified strains of either species in cholestasis research that could be used for a detailed evaluation, especially to define species'specific differences in hepatobiliary remodelling after tBDT.

## Recommendation when to select mice and when to select rats

Our results support the obvious choice of rats for surgical questions (e.g., repeated surgical intervention within one project) and mice for investigating molecular mechanism. Delicate hepatobiliary surgery is easier in rats and surgical procedures are better tolerated [12]. Our intention was and is not to discourage researchers from projects including hepatobiliary microsurgery in mice. Since the anatomy in mice demands highly precise knowledge and skills of hepatobiliary microsurgery, on should not undervalue the influence of the surgeon/ surgical part on the final results besides species'specific differences. Whereas, projects with genetically modified mice strains clearly benefit from the broad spectrum of further and specialised molecular analyses [1–3].

In summary, the key differences between rats and mice are their significantly different severity and dynamics of histological alterations and their different susceptibility to stress and injury (e.g., surgery, anaesthesia, body weight recovery, blood loss). These differences can gain important influence on results and success of projects, especially when the timing of special analyses is the most relevant issue. Key features of the molecular findings in mice might be confirmed in the rat model for an eventual species-specific effect. Furthermore, the knowledge of species-specific alterations can help minimising misleading data and unneeded usage of animals.

## Conclusion

The key difference between rats and mice is the severity and dynamics of histological alterations. In view of these differences, simple translation of the results obtained in mice on the situation in rats or even humans should be at least well considered.

## Acknowledgments

The authors would like to express their gratefulness to the excellent support of the technicians: Mrs. I. Jank for organisation of the office work, Mrs. St. Lange for her animal care, J. Schrimpf for measuring the blood samples, Mrs. E. Oswald and St. Lange for preparation of the histological slides and K. Schulze for staining and scanning of all the slides.

## Author Contributions

**Conceptualization:** Beate Richter.

**Data curation:** Beate Richter, Constanze Sänger, Franziska Mussbach.

**Formal analysis:** Beate Richter, Constanze Sänger, Franziska Mussbach.

**Funding acquisition:** Beate Richter, Utz Settmacher, Uta Dahmen.

**Investigation:** Beate Richter, Constanze Sänger.

**Methodology:** Beate Richter.

**Project administration:** Beate Richter.

**Resources:** Hubert Scheuerlein, Utz Settmacher, Uta Dahmen.

**Software:** Uta Dahmen.

**Supervision:** Hubert Scheuerlein, Uta Dahmen.

**Validation:** Beate Richter.

**Visualization:** Beate Richter.

**Writing – original draft:** Beate Richter.

**Writing – review & editing:** Beate Richter, Constanze Sänger, Franziska Mussbach, Hubert Scheuerlein, Utz Settmacher, Uta Dahmen.

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
