## [Decision Letter · Decision Letter 0]

9 Mar 2022

PONE-D-21-36588Species specific morphological alterations in liver tissue after biliary occlusion in rat and mouse: Similar but differentPLOS ONE

Dear Dr. Richter,

Thank you for submitting your manuscript to PLOS ONE. After careful consideration, we feel that it has merit but does not fully meet PLOS ONE’s publication criteria as it currently stands. Therefore, we invite you to submit a revised version of the manuscript that addresses the points raised during the review process.

I would like to sincerely apologise for the delay you have incurred with your submission. It has been exceptionally difficult to secure reviewers to evaluate your study. We have now received two completed reviews; their comments are available below. The reviewers have raised significant scientific concerns about the study that need to be addressed in a revision.

Please revise the manuscript to address all the reviewer's comments in a point-by-point response in order to ensure it is meeting the journal's publication criteria. Please note that the revised manuscript will need to undergo further review, we thus cannot at this point anticipate the outcome of the evaluation process.

We look forward to receiving your revised manuscript.

Kind regards,

Miquel Vall-llosera Camps

Senior Editor

PLOS ONE

Journal Requirements:

Reviewers' comments:

Reviewer's Responses to Questions

**Comments to the Author**

1. Is the manuscript technically sound, and do the data support the conclusions?

Reviewer #1: Yes

Reviewer #2: Partly

2. Has the statistical analysis been performed appropriately and rigorously? 

Reviewer #1: Yes

Reviewer #2: Yes

3. Have the authors made all data underlying the findings in their manuscript fully available?

Reviewer #1: No

Reviewer #2: Yes

4. Is the manuscript presented in an intelligible fashion and written in standard English?

Reviewer #1: Yes

Reviewer #2: No

5. Review Comments to the Author

Reviewer #1: Overall, the article reads well and contains good clear data to support the conclusions. The conclusions themselves are not overly exciting, but may be of practical use to researchers in the field. The one concern I have is whether strain-specific differences may exist, which is not explored in this study. Perhaps the authors should comment more on this in the discussion.

A major point is the word choice. There are several instances where poor word choice makes the resulting statement unclear. Specifically:

- In the abstract, the phrase “only tiny bile infarcts than mice” is grammatically incorrect. Use “only tiny bile infarcts compared to mice” instead.

- In the introduction, the phrase “rational… reasons” is redundant. Do you mean practical? Also, it is not clear what you mean by “for the abundance of reagents”. Are you trying to say that mice require lower volumes or amounts of reagents?

- In results, rephrase "albumin as parameters" to "serum albumin as a parameter". In the legend of Figure 1 where you say the mice did not "overreach" their weights, change it to "exceed". This is a better choice of words. Likewise, hepatocyte "compartment" is an inaccurate term. Use "hepatocyte mass" or "hepatocellular component" instead

A second major point is the lack of statistical comparison between groups. It is important to know if the observed differences are statistically significant. You should be using non-parametric tests based on your numbers.

A few minor points:

- I don't think that 100% vs. 92% survival is really a substantial difference, especially since it is much higher than other studies are reporting.

- It is unnecessary and somewhat distracting to bold and underling mice and rats.

- You don't need to put quotes around "biliary infarcts" every time, since it's a term you chosen to use.

- It's not clear what is meant by "portal fields" and "biliary proliferates". Do you mean periportal? Cholangiocyte proliferation?

Reviewer #2: This study aimed to demonstrate the suitability of using rats or mice for experimental occlusive cholestasis. Authors show that the main difference between both species are the severity and kinetics of histological alterations, and propose that septic issues are the most important factors impacting in mice survival. Although the finding might be of relevance, several issues still need to addressed before further steps.

The presentation of the manuscript is inadequate, authors repeat the figure legends in the manuscript and in the figures section; the words: rats and mice should not be either highlighted or underlined. Authors excessively use parenthesis throughout the manuscript; for example, in the abstract, conclusion section is too confused since it is not fluently written. Authors should avoid parenthesis and instead, they should rewrite the conclusion. In introduction: Mostly, rodents (rat, mouse) were used, because of rational and scientific reasons. It should be: Mostly, rodents either rats or mice HAVE BEEN used, because of rational and scientific reasons.

What does stand for 28“d” in the following sentence? At five time points (1, 3, 7, 14, 28d, n=5/time point) after tBDT,

According to The International System of Units (SI), hours and other units must be indicated as symbol (h), not as abbreviation (hrs); moreover, they should be separated from number; for example: 0.005 mg/kg instead of 0.005mg/kg, and so on and on. Authors should carefully review the whole manuscript.

Supplementary Tables were not included in the manuscript or review.

The manuscript contains a number of grammar and typo issues that need to be carefully reviewed; for example, the following sentence in result section, at page 6, states: “The rats showed AN maximal weight loss…”; but it must be: “The rats showed A maximal weight loss…”.

All abbreviations shown in figure 2 must be defined in the legend of the figure. For example, what does stand for CC, Hep, etc?

Figure legends contain excessive explanation which should be included in result section. Figure legends should contain a brief description about what images are showing but not a full explanation.

6. PLOS authors have the option to publish the peer review history of their article (what does this mean?). If published, this will include your full peer review and any attached files.

Reviewer #1: No

Reviewer #2: No

---

## [Author Response · Author response to Decision Letter 0]

11 May 2022

Reviewer #1: Overall, the article reads well and contains good clear data to support the conclusions. The conclusions themselves are not overly exciting, but may be of practical use to researchers in the field. The one concern I have is whether strain-specific differences may exist, which is not explored in this study. Perhaps the authors should comment more on this in the discussion.

A major point is the word choice. There are several instances where poor word choice makes the resulting statement unclear. Specifically:

- In the abstract, the phrase “only tiny bile infarcts than mice” is grammatically incorrect. Use “only tiny bile infarcts compared to mice” instead.

- In the introduction, the phrase “rational… reasons” is redundant. Do you mean practical? Also, it is not clear what you mean by “for the abundance of reagents”. Are you trying to say that mice require lower volumes or amounts of reagents

- In results, rephrase "albumin as parameters" to "serum albumin as a parameter". In the legend of Figure 1 where you say the mice did not "overreach" their weights, change it to "exceed". This is a better choice of words. Likewise, hepatocyte "compartment" is an inaccurate term. Use "hepatocyte mass" or "hepatocellular component" instead

We thank you very much for your important comments and references. We changed the wording and screened the manuscript for unnecessary or misleading description. We do very appreciate your helping suggestions.

A second major point is the lack of statistical comparison between groups. It is important to know if the observed differences are statistically significant. You should be using non-parametric tests based on your numbers.

We included statistical comparison of the groups. We marked the significances differences in the tables and figures. 

A few minor points:

- I don't think that 100% vs. 92% survival is really a substantial difference, especially since it is much higher than other studies are reporting. 

We agree with you that the difference between the survival rates is not very impressive. Since only one operative intervention (the biliary occlusion) led to a different mortality in rats and mice, we do think that this small numeric difference makes and describes a strong species specific distinction. Today we cannot explain these different survival rates. 

- It is unnecessary and somewhat distracting to bold and underling mice and rats.

We want to apologize for the distracting character of the visual marking of rats and mice within the description of the results. By using the visual marking, we wanted to simplify the distinction between rats and mice. We have withdrawn this labelling according to your recommendations. 

- You don't need to put quotes around "biliary infarcts" every time, since it's a term you chosen to use.

We apologize for using to many “quotations”, we have eliminated these and changed the writing according to your recommendations.

- It's not clear what is meant by "portal fields" and "biliary proliferates". Do you mean periportal? Cholangiocyte proliferation?

We apologize for the non-intended confusion by our wording. We do distinct between the ductular reaction in the periportal area including the portal fields/tract and the biliary proliferates in the hepatocellular compartment as description for cholangiocytes` proliferation due to ductular reaction at two different locations in the liver architecture. We use these different terms in order to simplify the differentiation between both intrahepatic localisation of cholangiocytes` proliferation (CC). Therefore, we use portal fields as “abbreviation” for the periportal CC and biliary proliferates in the hepatocellular compartment. We did delete the term extraportal in the manuscript.

Reviewer #2: This study aimed to demonstrate the suitability of using rats or mice for experimental occlusive cholestasis. Authors show that the main difference between both species are the severity and kinetics of histological alterations, and propose that septic issues are the most important factors impacting in mice survival. Although the finding might be of relevance, several issues still need to addressed before further steps.

The presentation of the manuscript is inadequate, authors repeat the figure legends in the manuscript and in the figures section; the words: rats and mice should not be either highlighted or underlined. Authors excessively use parenthesis throughout the manuscript; for example, in the abstract, conclusion section is too confused since it is not fluently written. Authors should avoid parenthesis and instead, they should rewrite the conclusion. In introduction: Mostly, rodents (rat, mouse) were used, because of rational and scientific reasons. It should be: Mostly, rodents either rats or mice HAVE BEEN used, because of rational and scientific reasons.

We do appreciate your important recommendations. We have changed the manuscript according to your suggestions. We have marked all changes with yellow.

What does stand for 28“d” in the following sentence? At five time points (1, 3, 7, 14, 28d, n=5/time point) after tBDT,

We used the international abbreviation d for days. We changed the sentence and the description for time intervals according to your recommendations.

According to The International System of Units (SI), hours and other units must be indicated as symbol (h), not as abbreviation (hrs); moreover, they should be separated from number; for example: 0.005 mg/kg instead of 0.005mg/kg, and so on and on. Authors should carefully review the whole manuscript. Supplementary Tables were not included in the manuscript or review.

The manuscript contains a number of grammar and typo issues that need to be carefully reviewed; for example, the following sentence in result section, at page 6, states: “The rats showed AN maximal weight loss…”; but it must be: “The rats showed A maximal weight loss…”.

We want to thank you very much for your important comments. We changed the manuscript and the writing according to your recommendations. We did upload the tables within the submission procedure. However, we included the tables in the revised manuscript and additionally submitted the tables in extra files as the tables demand extra space due to their formatting. We are very sorry that you have had no access to the single data of our study.

All abbreviations shown in figure 2 must be defined in the legend of the figure. For example, what does stand for CC, Hep, etc?

We apologize for the missing explanation of the abbreviations we used. We have included the explanations as you recommended. (CC stands for cholangiocellular, Hep for hepatocellular). 

Figure legends contain excessive explanation which should be included in result section. Figure legends should contain a brief description about what images are showing but not a full explanation.

We apologize for the non-intended redundance. We shortened the figure legends according to your comments.

---

## [Decision Letter · Decision Letter 1]

24 May 2022

PONE-D-21-36588R1Species specific morphological alterations in liver tissue after biliary occlusion in rat and mouse: Similar but differentPLOS ONE

Dear Dr. Richter,

Thank you for submitting your manuscript to PLOS ONE. After careful consideration, we feel that it has merit but does not fully meet PLOS ONE’s publication criteria as it currently stands. Therefore, we invite you to submit a revised version of the manuscript that addresses the points raised during the review process.

As reported below, these points mainly regard vocabulary, styles and in some cases reformulation of sentences. The modifications could thus be rapidly performed. Do not forget to address a point-by-point response to allow a quick view of the modifications.

We look forward to receiving your revised manuscript.

Kind regards,

Jean-Marc A Lobaccaro, PhD

Academic Editor

PLOS ONE

Journal Requirements:

Reviewers' comments:

Reviewer's Responses to Questions

**Comments to the Author**

1. If the authors have adequately addressed your comments raised in a previous round of review and you feel that this manuscript is now acceptable for publication, you may indicate that here to bypass the “Comments to the Author” section, enter your conflict of interest statement in the “Confidential to Editor” section, and submit your "Accept" recommendation.

Reviewer #1: (No Response)

Reviewer #2: All comments have been addressed

2. Is the manuscript technically sound, and do the data support the conclusions?

Reviewer #1: Partly

Reviewer #2: Yes

3. Has the statistical analysis been performed appropriately and rigorously? 

Reviewer #1: Yes

Reviewer #2: Yes

4. Have the authors made all data underlying the findings in their manuscript fully available?

Reviewer #1: No

Reviewer #2: Yes

5. Is the manuscript presented in an intelligible fashion and written in standard English?

Reviewer #1: Yes

Reviewer #2: Yes

6. Review Comments to the Author

Reviewer #1: Thank you for taking the time to address my comments. I still feel, however, that there are several issues that need to be addressed before publication.

Overall, there are still several instances of poor sentence structure and unusual or incorrect word choice. For instance, the sentence fragment "Moreover, transgenic mice strains facilitating further comparative molecular research." in the Introduction and the use of 'kinetic' as a noun throughout. I would recommend review by a native English speaker.

Section specific comments:

1. Abstract

- include p values

- write all abbreviations in full upon their first use (do not include an abbreviation if the word is only used once)

2. Methods

- state whether assessment was done in a blinded fashion

3. Results

- include p values in the text

- include data for rats and mice in the same tables, since you are making a comparison between them

- the appropriate abbreviations for alanine aminotransferase and aspartate aminotransferase are ALT and AST, respectively

- report the p value for mortality

- by my calculations it was 0.0740, which is generally not considered significant

- in light of this, you should report this a trend toward increased mortality in mice

- do you have baseline values for laboratory chemistry and weight (i.e., from untreated animals)?

- if so you should report these

- I have several problems with table 6 and I think it is best removed, all things considered

- the human column contains little useful information since you report most of the characteristics are dependent on the underlying disease state

- furthermore, the indication that 'genetical models' of humans are 'not applicable yet' is bizarre, as it would be highly unethical to ever create transgenic humans for research

- ratings in categories 'Tolerance to surgical stress' and 'Susceptibility to complications' appear highly subjective and it is unclear how you distinguish between 'low', 'moderate', and 'high'

- figure 6 provides all of the relevant information in this table

- if you think it is important to compare anatomical difference I would do so in a figure

4. Discussion

- conclusions are overstated

- the findings don't 'confirm' that rats are the best choice for surgical models

- they provide researchers with considerations when choosing between one or the other

- mice could still be used for surgical models, but researchers should understand that they are more susceptible to injury

- discuss limitations of your study

- the biggest factor that limits the applicability of your findings is the use of only male animals

- it seems that the impact of sex would be an important consideration when choosing an animal model for researchers in this field

- another is the unknown impact of strain on biliary injury

- it's hard to generalize these findings when you've only looked at one strain of each

Reviewer #2: The authors have properly addressed my suggestions.

I believe that the corrected version of the manuscript is now acceptable for further steps in the publication process by Plos One journal.

7. PLOS authors have the option to publish the peer review history of their article (what does this mean?). If published, this will include your full peer review and any attached files.

Reviewer #1: **Yes: **Joshua Hefler

Reviewer #2: No

---

## [Author Response · Author response to Decision Letter 1]

8 Jul 2022

Reviewer #1: Thank you for taking the time to address my comments. I still feel, however, that there are several issues that need to be addressed before publication.

Overall, there are still several instances of poor sentence structure and unusual or incorrect word choice. For instance, the sentence fragment "Moreover, transgenic mice strains facilitating further comparative molecular research." in the Introduction and the use of 'kinetic' as a noun throughout. I would recommend review by a native English speaker.

Section specific comments:

1. Abstract

- include p values

We have included the p-values in the abstract. Since we determined the dedicated values at five different time points, we mention only the significant values and their p-value. By doing so we want to avoid confusion between the values and to facilitate understanding of the data. We agree with you that a p-value below 0.07 is not associated with a “stronger” significance. Therefore, we distinct only between the p-value below or above/equal 0.05. 

- write all abbreviations in full upon their first use (do not include an abbreviation if the word is only used once)

We agree to use abbreviations after the word was written in full. We changed the manuscript.

2. Methods

- state whether assessment was done in a blinded fashion

We thank you for your interesting recommendation. We agree with you that a blinded assessment of data assures an objective evaluation of information. In particular in clinical studies the double-blinded assessment is an obligatory part of the study design. We agree with you that the clinical assessment of animals in experimental projects cannot be done in a blinded fashion. The histological slides were predominantly quantified by a software (batch analysis). However, the training of the software required always a human control. The blood test were analysed by an automated chemical device. However, the results were checked always for their validity by a human who was informed at least about time points, mostly also about the species. Since such quality controls in experimental projects are performed at regularly basis in our laboratory, we cannot state that the assessment was done in a blinded fashion.

3. Results

- include p values in the text

We have included the p-values in the results. 

- include data for rats and mice in the same tables, since you are making a comparison between them

We thank you for your important recommendation. We decided to present the data for rats and mice in separate tables facing diverse considerations: I.) we want to avoid confusion between data of rats and mice; II.) we want to facilitate comprehension and comparability of the huge amount of data. Therefore, we like to adhere to the presentation of the data of rats and mice in separate tables.

- the appropriate abbreviations for alanine aminotransferase and aspartate aminotransferase are ALT and AST, respectively

We thank you for your interesting request. We agree with you, that ALT and ALAT are equivalent abbreviations for alanine aminotransferase (as well as AST and ASAT are for aspartate aminotransferase) in the international literature. We have used ALAT and ASAT as abbreviations in numerous publications, and until now no international or national journal has ever asked for a correction into AST or ALT, respectively. We apologize for our decision of further use of ASAT and ALAT as abbreviations.

- report the p value for mortality

- by my calculations it was 0.0740, which is generally not considered significant

- in light of this, you should report this a trend toward increased mortality in mice

We agree with you that the differences in the survival rate were not significant with a p=0.070484. We included this value in the results section. In addition we included the description of significant differences.

- do you have baseline values for laboratory chemistry and weight (i.e., from untreated animals)?

- if so you should report these

Unfortunately, we do not have baseline values of untreated rats and mice on our time points of interests.

- I have several problems with table 6 and I think it is best removed, all things considered

- the human column contains little useful information since you report most of the characteristics are dependent on the underlying disease state

We agree with you that a comparison of results between different species is still challenging. By including Table 6 we want to facilitate and strengthen the understanding of important issues: I.) The differences in size (e.g., body weight, organ weight) and the differences in progression of weight/size due to tBDT (e.g., also the missing data in human in relation to the current literature); II.) the anatomical differences beyond the size issue (e.g., presence of a gall bladder, lobulation of liver); III.) the differences in stress resistance and susceptibility to complications (e.g., septic conditions, blood loss, anaesthesia, surgery); IV.) brief description of the characteristics/differences of the hepatobiliary remodelling after tBDT; V.) the massive differences in knowledge about the genetical background/ opportunities of genetical modification in relation to specific topics of the experimental cholestasis research. We included a column with human data from the literature to illustrate the differences in knowledge and the possible impossible comparison with human data (e.g., clinical studies, textbook data). We agree with you that in this case, the understanding of such a concentrated comparison of seven issues is best supported by the tablet format. 

- furthermore, the indication that 'genetical models' of humans are 'not applicable yet' is bizarre, as it would be highly unethical to ever create transgenic humans for research

We agree with your statement that it is unethical to create genetically altered humans for research. Obviously only you misinterpreted our sentence. We have changed the text in this line of the table 6. However, in medicine (e.g., cancer research, immunologic diseases research) we are about to gain more insights in the epigenetic background in animals (and humans) enabling or inhibiting or avoiding cancer and immunologic diseases. In experimental research, with the establishment of genetically modified species (predominantly mice; to a lesser extent rats) the scientific community built a network providing growing knowledge of altered pathways in relation to the genetical modification and diseases. 

- ratings in categories 'Tolerance to surgical stress' and 'Susceptibility to complications' appear highly subjective and it is unclear how you distinguish between 'low', 'moderate', and 'high. figure 6 provides all of the relevant information in this table, if you think it is important to compare anatomical difference I would do so in a figure

We agree with you that rating categories are at risk for smooth transition zones. However, in relation to our categories low vs. moderate vs. high concerning tolerance to surgical stress and susceptibility to complications we referred to the data of survival, body weight gain and stress score as already described in the section material and methods and section results. We included a description in the legend of the Table 6.

We agree with you that the Figure 6 presents the comparison of the differences in basic characteristic parameters (e.g., stress resistance/survival, characteristics of hepato-biliary remodelling after tBDT) for selection of the appropriate species (rat vs. mouse) in cholestasis research. Again, we agree with you that Table 6 and Figure 6 are complementary presentations of the results of this study and do support the fast understanding of species` specific differences. 

4. Discussion

- conclusions are overstated

- the findings don't 'confirm' that rats are the best choice for surgical models

- they provide researchers with considerations when choosing between one or the other

- mice could still be used for surgical models, but researchers should understand that they are more susceptible to injury

- discuss limitations of your study

We thank you for your interesting recommendations. We included a paragraph with discussion of the limitations of our study and changed the sentences according to your recommendations. 

The biggest factor that limits the applicability of your findings is the use of only male animals

- it seems that the impact of sex would be an important consideration when choosing an animal model for researchers in this field

We agree with you that the (still used) limitation on the male sex in experimental research maybe does harbour the risk of misleading data/ results. Since until now (nearly) all scientific data in cholestasis research raised from male animals, we decided to focus on male animals. We discussed this problem within the “limitations paragraph” in the discussion section.

- another is the unknown impact of strain on biliary injury

- it's hard to generalize these findings when you've only looked at one strain of each

We agree with you that every limitation of a study design harbour the risk of challenging results. In order to avoid a never ending project, we used only the male animals of one strain without genetical modifications per species and limited the study design on three endpoints. The strains were inbred C57BL/6N mice and inbred Lewis rats [2,28,34]. The endpoints were survival, stress resilience (e.g., body weight gain, stress score) and the hepato-biliary remodelling after tBDT. We used the mostly used strain of either species and only the male animals according to the current literature. Furthermore, we have not found equivalent genetically altered strains of either species in cholestasis research that could be used for a detailed evaluation, especially to define species` specific differences in hepatobiliary remodelling after tBDT. Therefore, we refrained from including genetically modified strains of either species.

Reviewer #2: The authors have properly addressed my suggestions.

I believe that the corrected version of the manuscript is now acceptable for further steps in the publication process by Plos One journal.

We thank you very much for your constructive comments.

---

## [Editor Report · Decision Letter 2]

12 Jul 2022

Species specific morphological alterations in liver tissue after biliary occlusion in rat and mouse: Similar but different

PONE-D-21-36588R2

Dear Dr. Richter,

We’re pleased to inform you that your manuscript has been judged scientifically suitable for publication and will be formally accepted for publication once it meets all outstanding technical requirements.

Kind regards,

Jean-Marc A Lobaccaro, PhD

Academic Editor

PLOS ONE
---

## [Editor Report · Acceptance letter]

14 Jul 2022

PONE-D-21-36588R2 

Species specific morphological alterations in liver tissue after biliary occlusion in rat and mouse: Similar but different 

Dear Dr. Richter:

I'm pleased to inform you that your manuscript has been deemed suitable for publication in PLOS ONE. Congratulations! Your manuscript is now with our production department. 

Kind regards, 

on behalf of

Dr. Jean-Marc A Lobaccaro 

Academic Editor

PLOS ONE